# Real-Time PCR-Based Screening for Homozygous *SMN2* Deletion Using Residual Dried Blood Spots

**DOI:** 10.3390/genes14122159

**Published:** 2023-11-29

**Authors:** Yoshihiro Bouike, Makoto Sakima, Yuya Taninishi, Takanori Matsutani, Yoriko Noguchi, Ryosuke Bo, Hiroyuki Awano, Hisahide Nishio

**Affiliations:** 1Faculty of Nutrition, Kobe Gakuin University, 518 Arise, Ikawadani-cho, Nishi-ku, Kobe 651-2180, Japan; bouike@nutr.kobegakuin.ac.jp (Y.B.); n8yemx03@s.kobegakuin.ac.jp (M.S.); n8xqww04@s.kobegakuin.ac.jp (Y.T.); 2Division of Physiology, Shinko Hospital, 1-4-47 Wakinohama-cho, Chuo-ku, Kobe 651-0072, Japan; t-matsutani@shinkohp.or.jp; 3Department of Clinical Laboratory, Kobe University Hospital, 7-5-1 Kusunoki-cho, Chuo-ku, Kobe 650-0017, Japan; ynoguchi@med.kobe-u.ac.jp; 4Department of Pediatrics, Kobe University Graduate School of Medicine, 7-5-1 Kusunoki-cho, Chuo-ku, Kobe 650-0017, Japan; ryobo@med.kobe-u.ac.jp; 5Organization for Research Initiative and Promotion, Research Initiative Center, Tottori University, 86 Nishi-cho, Yonago 683-8503, Japan; awano@tottori-u.ac.jp; 6Department of Occupational Therapy, Faculty of Rehabilitation, Kobe Gakuin University, 518 Arise, Ikawadani-cho, Nishi-ku, Kobe 651-2180, Japan

**Keywords:** spinal muscular atrophy, *SMN1*, *SMN2*, motor neuron diseases, real-time PCR, dried blood spot

## Abstract

The survival motor neuron 2 (*SMN2*) gene is a recognized modifier gene of spinal muscular atrophy (SMA). However, our knowledge about the role of *SMN2*—other than its modification of SMA phenotypes—is very limited. Discussions regarding the relationship between homozygous *SMN2* deletion and motor neuron diseases, including amyotrophic lateral sclerosis, have been mainly based on retrospective epidemiological studies of the diseases, and the precise relationship remains inconclusive. In the present study, we first estimated that the frequency of homozygous *SMN2* deletion was ~1 in 20 in Japan. We then established a real-time polymerase chain reaction (PCR)-based screening method using residual dried blood spots to identify infants with homozygous *SMN2* deletion. This method can be applied to a future prospective cohort study to clarify the relationship between homozygous *SMN2* deletion and motor neuron diseases. In our real-time PCR experiment, both PCR (low annealing temperatures) and blood (high hematocrit values and low white blood cell counts) conditions were associated with incorrect results (i.e., false negatives and positives). Together, our findings not only help to elucidate the role of *SMN2*, but also aid in our understanding of the pitfalls of current SMA newborn screening programs for detecting homozygous *SMN1* deletions.

## 1. Introduction

The survival motor neuron 2 (*SMN2*) gene is a modifier gene of spinal muscular atrophy (SMA); a lower *SMN2* copy number is associated with increased disease severity [1]. SMA is one of the more common lower motor neuron diseases with autosomal recessive inheritance. It is characterized by the degeneration of the anterior horn cells of the spinal cord, leading to proximal muscle atrophy and weakness [1]. 

In 1995, *SMN1*, located on human chromosome 5q13, was identified as the causative gene for SMA [2]. Chromosome 5q13 has a large duplication region, and two homologous genes (i.e., paralogs) are aligned on the telomeric and centromeric sides. *SMN1* is on the telomeric side, and *SMN2* is on the centromeric side. Lefebvre et al. reported the absence of *SMN1* (i.e., homozygous *SMN1* deletions) in more than 95% of SMA patients; intragenic mutations in *SMN1* (i.e., small mutations in *SMN1*) were identified in the remaining patients [2]. However, an absence of *SMN2* was found in 5% of control individuals but in none of the patients with SMA [2]. No patients with a complete absence of both *SMN1* and *SMN2* have yet been reported [1].

*SMN1* and *SMN2* are almost identical, with the exception of several single nucleotides [2,3,4]. The cytosine nucleotide (C) at the 6th position of exon 7 of *SMN1* causes exon 7 inclusion in all *SMN1* mRNA species (i.e., full-length *SMN1* mRNA). By contrast, the thymine nucleotide (T) at the 6th position of exon 7 of *SMN2* causes exon skipping in a large portion of *SMN2* mRNA (i.e., △7 *SMN2* mRNA) [5,6]. Notably, a small portion of full-length *SMN2* mRNA, including exon 7, may play a role in improving SMA symptoms [1], although it has been reported that the protein product of △7 SMN2 mRNA (i.e., △7 SMN2 protein) may increase the amount of functional SMN complex that is present in the motor neurons of SMA model animals [7]. In any case, *SMN2* works to compensate for the loss of *SMN1*, at least to some extent. 

Other than as a modifier of SMA, the role of *SMN2* remains unclear. There are some reports of a relationship between homozygous *SMN2* deletion and SMA with distal muscle involvement. For example, in 2001, Srivastava et al. reported a patient with homozygous *SMN2* deletion and distal muscle involvement, suggesting that *SMN2* might confer additional disease susceptibility in a select subset of SMA patients [8]. However, this report was criticized because it appeared to ignore the relatively high frequency of homozygous *SMN2* deletions in the general population [9]. Nonetheless, in 2012, Liping et al. reported a similar case with homozygous SMN2 deletion and distal muscle involvement [10].

The relationship between homozygous *SMN2* deletion and amyotrophic lateral sclerosis (ALS) has also been controversial. In 1998, Moulard et al. identified homozygous deletions of *SMN2* in 36% of individuals with sporadic adult-onset lower motor neuron disease, but in only 6.2% of individuals with sporadic ALS and in 1.5% of individuals with familial ALS [11]. However, the lower motor neuron disease patients in their study were later recognized as having the lower motor neuron form of ALS [12]. Subsequently, the relationship between *SMN2* and ALS has often been discussed [13,14,15,16,17]; some researchers have concluded that homozygous *SMN2* deletion may be a risk factor for ALS [14,16], while others have stated that it may be a protective factor for the disease [15]. To further complicate matters, a third group of researchers has insisted that homozygous *SMN2* deletion is not related to survival or respiratory decline in ALS patients [17].

The quarter-century discussion about the relationship between homozygous *SMN2* deletion and motor neuron diseases has been mainly based on retrospective epidemiological studies of the diseases, and the data remain inconclusive. Notably, there have been no reports of prospective cohort studies of people with homozygous *SMN2* deletion to date. Prospective cohort studies may allow for a more definite conclusion regarding the relationship between homozygous *SMN2* deletion and motor neuron diseases. Unfortunately, cohort studies require large amounts of funding and effort, and are unlikely to be readily available. However, we believe that establishing a methodology of real-time polymerase chain reaction (PCR) with residual dried blood spot (DBS) samples would be worthwhile for its use in future cohort studies. 

The establishment of such a methodology would also be of great benefit to the current newborn screening for SMA (SMA-NBS), which detects homozygous *SMN1* deletion and is now being implemented around the world. The incidence of SMA is approximately 1 in 10,000 to 20,000 live births [18] and, as mentioned earlier, homozygous *SMN1* deletions have been identified in more than 95% of SMA patients. Currently, the main methodology of SMA-NBS involves the detection of homozygous *SMN1* deletion with real-time PCR using DBSs. We are able to assume that the factors that affect the results of real-time PCR using DBSs are common to systems used to screen the homozygous deletion of *SMN1* or *SMN2*. 

In the present study, we first estimated the frequency of homozygous *SMN2* deletion in Japan using PCR restriction fragment length polymorphism (PCR-RFLP) with DBSs. We then established a real-time PCR-based screening method using residual DBSs to detect infants with homozygous *SMN2* deletion. In the future, this method can be applied to a future prospective cohort study to clarify the relationship between homozygous *SMN2* deletion and motor neuron diseases. We also studied the perturbation factors that may cause incorrect results in the screening system for homozygous *SMN2* deletion; these findings will be helpful for understanding the potential pitfalls of the current SMA-NBS programs used to detect homozygous *SMN1* deletion.

## 2. Materials and Methods

### 2.1. Residual DBS Samples and Ethics Committee Approvals

In the current study, 300 residual DBS samples from *SMN1*-retained infants were analyzed. They were randomly selected from residual samples that had been collected from 4157 newborn infants in Japan between January 2018 and April 2019 for our previous pilot study [19]. All 4157 newborn infants were confirmed to carry at least one copy of *SMN1*. Although the samples had been anonymized, this study was made public through an opt-out procedure. The study was approved by the Ethics Committee of Kobe University Graduate School of Medicine (reference B230027, approved on 14 June 2023) and the Ethics Committee of Kobe Gakuin University (regarding this study with residual DBS samples conducted in Kobe Gakuin University: reference 23-02, approved on 16 October 2023). Both the previous pilot investigation and the present study were conducted in accordance with the World Medical Association Declaration of Helsinki.

### 2.2. PCR-RFLP

PCR-RFLP was performed according to the method reported by van der Steege et al. [20]. Briefly, a punched circle (1.2 mm in diameter) from the DBS card was placed directly into a PCR mixture containing 1 unit of DNA polymerase KOD FX Neo™ (Toyobo, Osaka, Japan) to a final volume of 50 μL (Appendix A). Primer sequences are shown in Table 1.

To amplify exon 7 of *SMN1/SMN2* in the DBS samples, conventional PCR experiments were performed using a Mastercycler^®^ Nexus (Eppendorf SE, Tokyo, Japan). The PCR conditions were as follows: (1) an initial denaturation at 94 °C for 7 min, (2) 30 cycles of denaturation at 94 °C for 1 min, annealing at 56 °C for 1 min, and extension at 72 °C for 1 min, and (3) a final extension at 72 °C for 7 min (Appendix A).

Thereafter, restriction enzyme treatment was performed on the amplified products using the restriction enzyme Dra I (Takara Bio Inc., Shiga, Japan) at 37 °C for 12 h (Appendix A). An aliquot of the digested product was then electrophoresed on a 4% agarose gel in 1× Tris/borate/ethylenediaminetetraacetic acid buffer and visualized using Midori Green Direct Stain (Nippon Genetics, Tokyo, Japan). Figure 1 shows a diagram illustrating the primer positions and pre- and post-digestion products.

### 2.3. Real-Time PCR

A punched circle (1.2 mm in diameter) from the DBS card was placed directly into the PCR mixture containing 1 unit of DNA polymerase KOD FX Neo, intercalating fluorescent dye EvaGreen^®^ (Biotium, Hayward, CA, USA), and ROX Reference Dye (Thermo Fisher Scientific Inc., Waltham, MA, USA) to a final volume of 25 μL (Appendix A). The ROX was used to correct the well-to-well variation caused by pipetting inaccuracies and fluorescence fluctuations. The primer sequences are shown in Table 1.

To amplify exon 7 of *SMN2* in the DBS samples, real-time PCR experiments were performed using the StepOne™ Real-Time PCR System (Thermo Fisher Scientific Inc.). The PCR conditions for the 300 samples were as follows: (1) an initial denaturation at 94 °C for 7 min, (2) 45 cycles of denaturation at 94 °C for 1 min, annealing at 56 °C, 58 °C, 60 °C, or 62 °C for 1 min, and extension at 72 °C for 1 min, and (3) a final extension at 72 °C for 7 min (Appendix A). Figure 2 shows a diagram illustrating the primer positions used in the real-time analysis in this study.

### 2.4. Analysis of Perturbation Factors in the DBS Samples

For the purpose of analyzing perturbation factors in the DBS samples, we collected fresh ethylenediaminetetraacetic acid-anticoagulated blood from a healthy volunteer (Appendix A). We confirmed that *SMN2* was retained in the blood prior to use. 

To investigate the effects of red blood cells on real-time PCR amplification, three types of blood with different hematocrit values (40%, 60%, and 80%) were prepared by mixing concentrated red blood cells and plasma. The white blood cell (WBC) counts of these artificial bloods were roughly similar (Appendix A). 

To explore the effects of WBCs on real-time PCR amplification, three types of blood with different WBC counts were prepared by adding concentrated WBCs to the artificial blood with 50% hematocrit; the WBC counts of the artificial bloods were 300, 700 and 4100/μL (Appendix A). 

The complete blood counts of all artificial blood prepared in this study were performed using a Sysmex KX-21 (Sysmex Corporation, Hyogo, Japan). The precise data of the artificial bloods are shown in Appendix A. Then, we prepared DBS samples on the FTA^®^ Elute Cards using these artificial bloods.

The PCR conditions for the DBS samples with artificial blood were as follows: (1) an initial denaturation at 94 °C for 7 min, (2) 55 cycles of denaturation at 94 °C for 1 min, annealing at 62 °C for 1 min, and extension at 72 °C for 1 min, and (3) a final extension at 72 °C for 7 min (Appendix A). We repeated the whole procedure with the same DBS sample five times.

### 2.5. Statistical Analyses

After creating contingency tables, the sensitivity and specificity values of homozygous SMN2 deletion under different PCR conditions (different annealing temperatures) were determined using a Microsoft Excel add-in software, BellCurve for Excel (Social Survey Research Information Co., Ltd. Tokyo, Japan). The sensitivity, specificity, and their 95% confidence intervals (95% CIs) were calculated using the same software. Welch’s *t*-test was performed using Microsoft Excel to compare the Ct values between two samples with different hematocrit values and between samples with different WBC counts. A *p*-value of <0.05 was considered significant in all statistical analyses.

## 3. Results

### 3.1. Detection of SMN2 Deletion by PCR-RFLP

To confirm the presence or absence of *SMN2* in the 300 DBS samples, each DNA pre-amplification product was digested by incubating the samples overnight with Dra I. The pre-amplified products derived from *SMN2* had a Dra I restriction enzyme site (TTT AAA) and were digested by Dra I, whereas the pre-amplified products derived from *SMN1* did not have a Dra I site and were not digested.

As shown in Figure 1, the PCR fragment size of *SMN2* before Dra I digestion is 187 bp. Digestion with Dra I produced two fragments, 163 bp and 24 bp in size. Because the small 24 bp fragment is difficult to see clearly on agarose gels, previous studies, including van der Steege et al. [20] and Srivastava et al. [8], have used only the large 163 bp fragment as clear evidence for the presence of *SMN2*. Thus, we show only the large 163 bp fragment in the gel electrophoresis picture. 

Figure 3 shows the results of four representative DBS samples. In the post-digestion products of Samples 119 and 124, the presence of both *SMN1* (an upper band, 187 bp) and *SMN2* (a lower band, 163 bp) can be observed, whereas in the post-digestion products of Samples 108 and 111, the presence of *SMN1* (an upper band, 187 bp) and the absence of *SMN2* (no band) can be observed. We therefore determined that Samples 119 and 124 were true-negative for homozygous *SMN2* deletions, and that Samples 108 and 111 were true-positive for homozygous *SMN2* deletions.

In the PCR-RFLP assay, homozygous *SMN2* deletion was detected in 16 of the 300 samples. Based on these results, we determined that 16 samples were true-positive homozygous *SMN2* deletions.

The left and right bands of the fragment pairs show the pre- and post-digestion products, respectively. The post-digestion products indicate that Samples 119 and 124 were true-negative for homozygous *SMN2* deletions, whereas Samples 108 and 111 were true-positive for homozygous *SMN2* deletions.

### 3.2. Detection of SMN2 Deletion by Real-Time PCR

#### 3.2.1. Amplification Curves of DBS Samples with and without Homozygous SMN2 Deletion

We compared the results of real-time PCR using annealing temperatures of 56 °C, 58 °C, 60 °C, and 62 °C. In the present study, we determined the true-positive or true-negative status based on the results of the PCR-RFLP analysis. As shown in Figure 4, when real-time PCR was performed at annealing temperatures of 56 °C–60 °C, the amplification curves of DBS samples with and without homozygous SMN2 deletion were unable to be distinguished. However, at 62 °C, the amplification curves of DBS samples with and without homozygous SMN2 deletion were able to be clearly separated.

Samples 119 and 124 did not have homozygous *SMN2* deletions, whereas Samples 108 and 111 had homozygous *SMN2* deletions. When real-time PCR was performed at annealing temperatures of 56 °C–60 °C, the amplification curves of DBS samples with and without homozygous *SMN2* deletion were unable to be distinguished (A, B, C). However, at 62 °C, the amplification curves of DBS samples with and without homozygous *SMN2* deletion were able to be clearly distinguished (D). DBS, dried blood spot.

#### 3.2.2. False Positives and False Negatives at Different Annealing Temperatures

In the real-time PCR analysis at specific annealing temperatures, DBS samples that had amplification curves with a cycle threshold (Ct) value > the mean plus 2 or more standard deviations were defined as *SMN2*-lacking samples. Table 2 contains four contingency tables showing the results of PCR-RFLP and real-time PCR. At an annealing temperature of 64 °C, the amplification efficiency was extremely reduced in many DBS samples.

When real-time PCR was performed at an annealing temperature of 62 °C, the mean and SD of the Ct values of the DBS samples retaining *SMN2* (based on the data of PCR-RFLP, n = 284) were 33.02 and 2.43, respectively. On the other hand, the mean and SD of the Ct values of the DBS samples lacking *SMN2* (n = 16) were 40.26 and 1.83, respectively. Thus, we used 37.88 as the cutoff Ct value of mean plus 2SD. Here, the sensitivity was 93.8% (95% CI: 67.3%, 99.99%) and the specificity was 98.9% (95% CI: 96.84%, 99.77%). We judged 62 °C to be the best annealing temperature in the present study.

We also performed a sequencing analysis of a false negative sample. The sequencing results of the false-negative sample are shown in the Appendix A). This sample suggested that *SMN2*-specific primers can amplify *SMN1* sequences via the mis-annealing of the primers to the *SMN1* sequence under some conditions, including a low annealing temperature.

### 3.3. Perturbation Factors in DBS Samples

To investigate perturbation factors, we performed real-time PCR using the artificial blood with different hematocrit values or WBC counts (Figure 5). To examine the effects of the hemoglobin concentration on PCR amplification, we prepared three DBS samples using artificial bloods with different hematocrit values (40%, 60%, and 80%). As the hematocrit value increased, the Ct value also increased. Similarly, to investigate the effect of WBC counts, we prepared three DNA samples with different WBC counts (300, 700, and 4100/μL). As the WBC count decreased, the Ct value increased.

## 4. Discussion

### 4.1. Use of Residual DBS Specimens after Testing

In the present study, we used residual DBS samples from an earlier pilot study of newborn screening for *SMN1*-deleted SMA. Our findings indicate that molecular genetic testing using residual DBS samples is possible; this technique will be used in a future prospective cohort study to clarify the relationship between homozygous *SMN2* deletion and motor neuron diseases. 

A prospective cohort study starting from the neonatal period is likely to be very expensive and difficult to perform. In addition, many complex issues of privacy, security, and technological coordination will need to be resolved. However, the testing of residual DBS—if combined with information in public health registries—may be able to provide an integrated picture of the health of entire populations, starting from birth [22]. 

### 4.2. Establishment of the Real-Time PCR Screening Method

#### 4.2.1. Comparison between Intercalating Dye and Fluorescent Probe Methods 

There are two main methodologies employed for analysis using real-time PCR: intercalating dye and fluorescent probe methods. In the intercalating dye method, it is comparatively easy to find proper PCR conditions because the only factor to consider is the sequences of primers that anneal to the gene of interest. In addition, the intercalating dye is not expensive. The weakness of the intercalating dye method is that it cannot test multiple genes at the same time. 

On the contrary, in the fluorescent probe method, multiple genes are tested at the same time. However, it is difficult to find the proper conditions under which many primers and fluorescent probes work well in one tube. In addition, fluorescent probes are expensive. The more probes there are, the more it costs.

In this study, we adopted the intercalating dye method with EvaGreen^®^, because the target gene was only *SMN2*. However, determining the proper annealing temperature required trial and error, as shown in the Results section.

#### 4.2.2. PCR-RFLP Analysis Demonstrating a High Frequency of Homozygous *SMN2* Deletion

We performed PCR-RFLP analysis to detect DBS samples with homozygous *SMN2* deletion; this method of separating *SMN1* and *SMN2* was developed by van der Steege et al. [20]. 

Compared with real-time PCR, which is commonly used for genetic newborn screening, PCR-RFLP is laborious; the procedure involves the enzymatic digestion of PCR products, the preparation of agarose gels, and electrophoresis. However, this method is very robust and produces stable results because a sufficient quantity of PCR product can be obtained regardless of the quality and quantity of DNA from the original DBS samples, and the product can be completely digested with sufficient time. In addition, the amplified products can be checked according to the band size. This method can therefore be used to confirm the results obtained using real-time PCR [19]. In the current study, we determined true-positive and -negative results based on the PCR-RFLP analysis.

Our PCR-RFLP analysis revealed a frequency of homozygous *SMN2* deletion of ~1 in 20 in Japan (~5%, 16 of 300 samples). Compared with the reported frequency of homozygous *SMN1* deletion in Japan (1 in 20,000–25,000 [23]), the frequency of homozygous *SMN2* deletion appears much higher. Our findings are similar to those of the first report of SMN2 by Lefebvre et al., in which ~5% of normal, asymptomatic individuals were reported to lack both copies of *SMN2* [2].

#### 4.2.3. False-Negative Results Caused by Lower Annealing Temperatures

Real-time PCR is preferable to PCR-RFLP when handling a large number of samples. However, RT-PCR draws an amplification curve without considering the effects of nonspecific amplification products. Specific amplification is therefore very important for obtaining accurate results. In the present study, to identify the optimal PCR conditions for detecting samples with homozygous *SMN2* deletion, we first tested different annealing temperatures (56 °C, 58 °C, 60 °C, and 62 °C).

When an annealing temperature of 56–60 °C was used, we failed to detect many samples with homozygous *SMN2* deletion (Table 2). These samples were thus recognized as false-negative samples. We also identified a false-positive case that appeared to have no *SMN2* fragments. However, when an annealing temperature of 62 °C was used, we were able to detect 15 of the 16 samples with homozygous *SMN2* deletion, which means that the detection sensitivity of homozygous *SMN2* deletion was much improved under this condition (Table 2). Nonetheless, even with an annealing temperature of 62 °C, we failed to detect one sample with homozygous *SMN2* deletion; this sample was recognized as a false-negative sample. In addition, with an annealing temperature of 62 °C, we identified three false-positive cases that apparently carried homozygous *SMN2* deletions (Table 2).

Together, these findings suggest that even if allele-specific primers are used in the analysis, a low annealing temperature negates the allele specificity of the primers, thus producing false-negative results. To maintain the stringency of allele-specific primers, a relatively high annealing temperature is therefore necessary. However, it is important to note that higher annealing temperatures may increase the number of false-positive results, likely because they suppress the amplification of *SMN2* fragments in DBS samples. Furthermore, a relatively high annealing temperature was unable to completely prevent the occurrence of false-negative findings in our study. Further investigation is therefore necessary to identify the causes of incorrect results in DBS samples.

#### 4.2.4. False-Positive Results Caused by Improper DBS Preparation 

We next investigated the possible perturbation factors of DBS samples that may lead to incorrect results. The inhibitory effects of heparin on PCR amplification efficiency have been reported elsewhere [23]. In the present study, we assumed that hemoglobin and the DNA concentration may be perturbation factors; hematocrit values and WBCs count were used as their respective markers. 

In the current study, we performed PCR analysis without DNA extraction and purification (i.e., direct PCR analysis). Although blood compounds, including hemoglobin, are known to inhibit PCR [24], recent studies using PCR enhancer cocktails and inhibition-resistant polymerases have enabled direct PCR [25], which is often used to save time and reduce costs in routine analysis [24]. 

According to the manufacturer’s information, the DNA polymerase used in our study (KOD FX Neo) gives successful results during PCR amplification with blood samples. However, our data clearly showed that DBSs with higher hematocrit values resulted in higher Ct values than those with lower hematocrit values. This may be because hemoglobin caused the quenching of fluorescence from the double-stranded DNA-binding dye (i.e., intercalating dye) EvaGreen^®^ [26]. This finding suggests that DBS samples with higher hemoglobin concentrations may lead to a failure to detect copies of *SMN2* (i.e., false-positive results).

Our data also revealed that DBSs with lower WBC counts resulted in higher Ct values than those with higher WBC counts. This may be because smaller amounts of DNA were contained in DBSs with lower WBC counts. This finding suggests that DBS samples with lower WBC counts may also lead to a failure to detect copies of *SMN2* (i.e., false-positive results). 

Improper DBS sample preparation that leads to a high hemoglobin concentration or low WBC count should be avoided, in any case. However, we have often encountered DBS samples that are not appropriate for testing. When blood is caked, clotted, or layered onto the filter paper, it suggests that the DBS sample has a high hemoglobin concentration. When blood is of insufficient quantity for testing, it suggests the DBS sample has a low WBC count. It should therefore be noted that all of these conditions may cause false-positive results.

### 4.3. Prediction of Possible SMA-NBS Pitfalls

For the early diagnosis of SMA, an SMA-NBS program, which analyzes DBS samples using real-time PCR, has been implemented worldwide [27]. The current SMA-NBS is used to detect the homozygous deletion of SMN1, which is found in 95% of SMA patients [2]. 

A careful review of previous studies on SMA screening using DBSs revealed that a frequently used technique to detect *SMN1* deletion involves real-time PCR technology with a fluorescent hybridizing probe; the fluorescent probe binds to an *SMN1*-specific sequence [28,29] or to a common sequence between *SMN1* and *SMN2* [30]. In contrast, our laboratory has reported several SMA screening systems that use real-time PCR technology with no fluorescent hybridizing probe; we instead use intercalating dye [19,31].

We speculate that the technical issues of real-time PCR have not been fully investigated in each testing center, as mentioned in the Introduction section. Retesting is usually not performed if *SMN1* is determined to be present, meaning that false-negative cases may be missed. In our previous pilot study of SMA-NBS, we conducted a follow-up survey to check for missing false-negative cases, although no positive cases were found [19].

The lessons learned from the present study can be applied to SMA-NBS. Our data revealed that, depending on the PCR conditions, false-negative and -positive results may occur. Furthermore, depending on the conditions of the DBS samples, false-positive results may occur. Given that each SMA-NBS testing center has different equipment, it is necessary to explore the optimal PCR conditions for each center. In addition, the preparation of DBS specimens is critical; the training of hospital staff who handle newborns is therefore essential.

### 4.4. Limitations

Our genetic testing with a real-time PCR method using DBSs is just a screening of the disease. Genetic testing using DBSs sometimes may produce inconclusive results. Therefore, in the real clinical spots, it is necessary for us to refer the individuals who had inconclusive DBS samples to expert doctors for the exact diagnosis.

## 5. Conclusions

We determined that the frequency of homozygous SMN2 deletion was ~1 in 20 in Japan. We also established a real-time PCR-based screening method using residual DBSs to detect homozygous SMN2 deletion in infants; this method may be applied to a future prospective cohort study to clarify the relationship between homozygous SMN2 deletion and motor neuron diseases. In our real-time PCR experiments, both PCR (low annealing temperatures) and blood (high hematocrit values and low WBC counts) conditions were associated with incorrect results (false negatives and positives). Together, these findings not only help to elucidate the role of *SMN2*, but also aid in our understanding of the pitfalls of current SMA-NBS programs that detect homozygous *SMN1* deletions.

## Figures and Tables

**Figure 1 genes-14-02159-f001:**
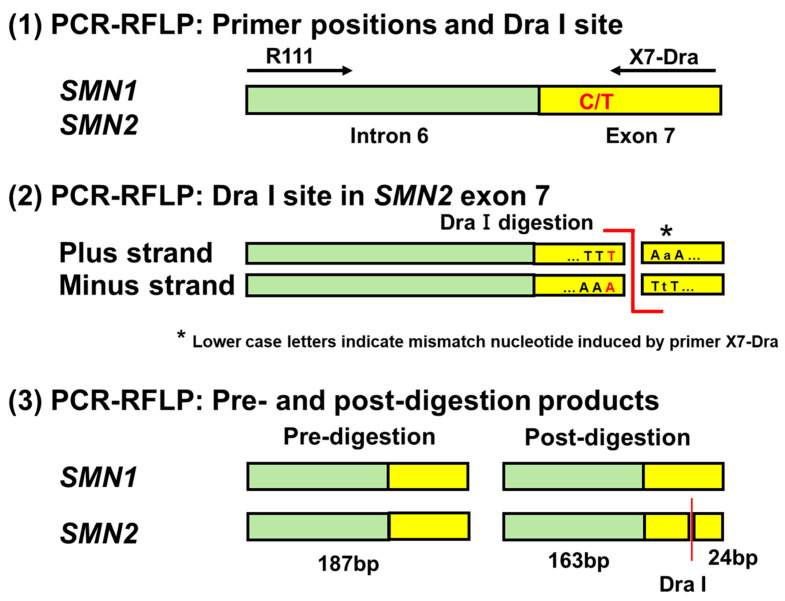
Primer positions and pre- and post-digestion products in PCR-RFLP analysis. Dra I treatment creates two products of 187 bp and 24 bp from *SMN2* fragment, but it does not lead to any change in *SMN1* fragment.

**Figure 2 genes-14-02159-f002:**
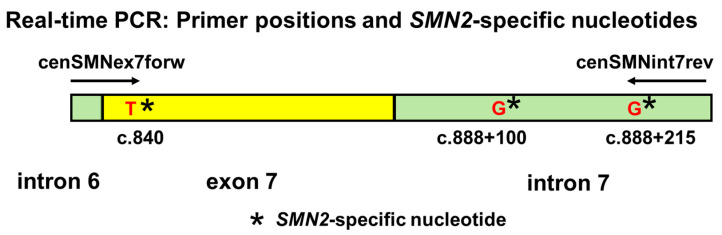
Primer positions and *SMN2*-specific nucleotides. *SMN2* amplification products contain three *SMN2*-specific nucleotides.

**Figure 3 genes-14-02159-f003:**
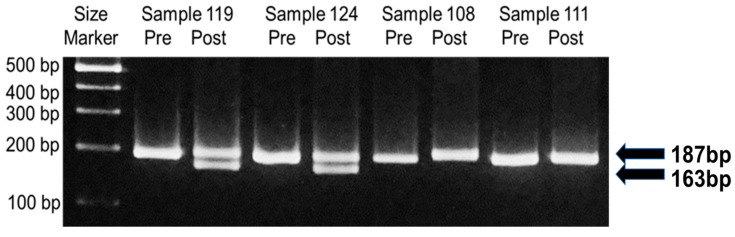
Detection of *SMN2* deletion using PCR-RFLP.

**Figure 4 genes-14-02159-f004:**
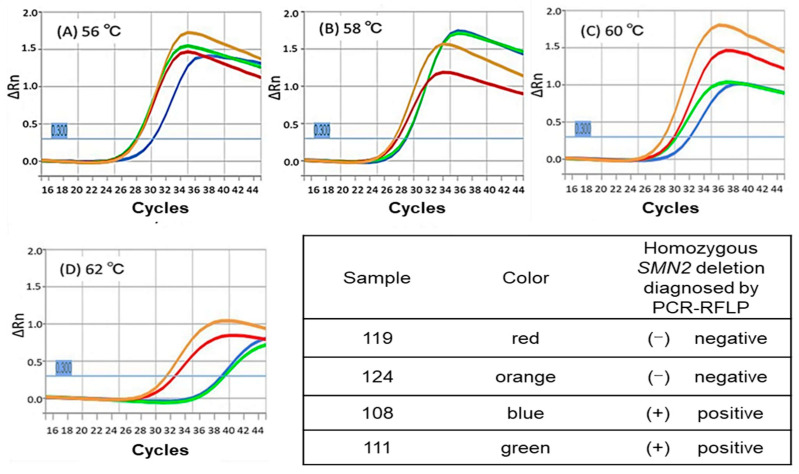
Amplification curves of DBS samples using real-time PCR.

**Figure 5 genes-14-02159-f005:**
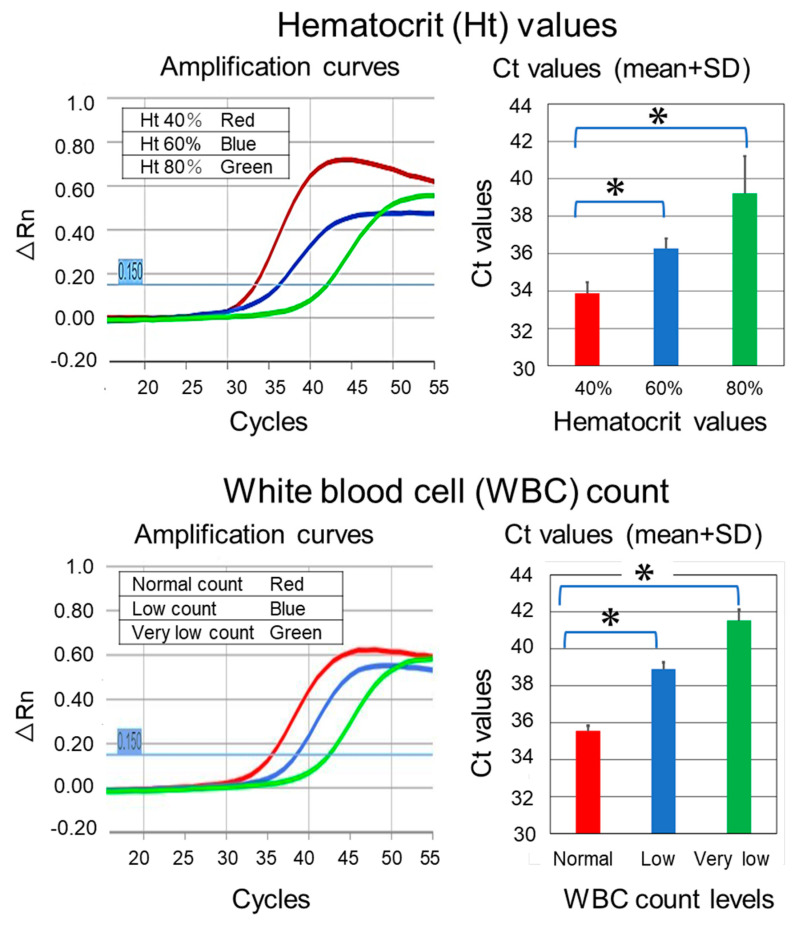
Effects of perturbation factors on amplification curves and Ct values. The upper panel summarizes the effects of different hematocrit values on amplification curves and Ct values. Higher hematocrit values appeared to inhibit the amplification of the target fragment, although they may also have quenched the fluorescence (see the Discussion for details). The lower panel summarizes the effects of different WBC counts on the amplification curves and Ct values. Lower WBC counts appeared to lower the amplification of the target fragment. “Normal count level”, “Low count level” and “Very low count level” refer to WBC counts of 4100, 700 and 300/µL, respectively. PCR experiments using the DBS samples with artificial blood were repeated five times. SD, standard deviation; WBC, white blood cell; * *p* < 0.01.

**Table 1 genes-14-02159-t001:** Nucleotide sequences of the primers used in this study.

	Primer Name	Sequences (5′→3′)	Ref.
PCR-RFLP	R111	AGA CTA TCA ACT TAA TTT CTG ATCA	[2]
	X7-Dra	CCT TCC TTC TTT TTG ATT TTG TtT	[20]
Real-time PCR	cenSMNex7forw	TTT ATT TTC CTT ACA GGG TTT TA	[21]
	cenSMNint7rev	GTG AAA GTA TGT TTC TTC CAC gCA	

Lowercase letters indicate mismatched nucleotides introduced to the primer sequences. PCR, polymerase chain reaction; RFLP, restriction fragment length polymorphism.

**Table 2 genes-14-02159-t002:** Sensitivity and specificity of real-time PCR at different annealing temperatures.

(A) Annealing temperature of 56 °C	PCR-RFLP	Total
Homozygous *SMN2* deletion (+)	Homozygous *SMN2* deletion (–)	
Real-time PCR	Homozygous *SMN2* deletion (+)	4	5	9
Homozygous *SMN2* deletion (–)	12	279	291
Total	16	284	300
Sensitivity: 25.00% [95% CI: 7.27%, 52.38%]
Specificity: 98.20% [95% CI: 95.87%, 99.41%]

(B) Annealing temperature of 58 °C	PCR-RFLP	Total
Homozygous *SMN2* deletion (+)	Homozygous *SMN2* deletion (–)	
Real-time PCR	Homozygous *SMN2* deletion (+)	2	5	7
Homozygous *SMN2* deletion (–)	14	279	293
Total	16	284	300
Sensitivity: 12.50% [95% CI: 1.55%, 38.35%]
Specificity: 98.20% [95% CI: 95.87%, 99.41%]

(C) Annealing temperature of 60 °C	PCR-RFLP	Total
Homozygous *SMN2* deletion (+)	Homozygous *SMN2* deletion (–)	
Real-time PCR	Homozygous *SMN2* deletion (+)	7	7	14
Homozygous *SMN2* deletion (–)	9	277	286
Total	16	284	300
Sensitivity: 43.75% [95% CI: 19.75%, 70.12%]
Specificity: 97.54% [95% CI: 94.92%, 99.05%]

(D) Annealing temperature of 62 °C	PCR-RFLP	Total
Homozygous *SMN2* deletion (+)	Homozygous *SMN2* deletion (–)	
Real-time PCR	Homozygous *SMN2* deletion (+)	15	3	18
Homozygous *SMN2* deletion (–)	1	281	282
Total	16	284	300
Sensitivity: 93.80% [95% CI: 67.30%, 99.99%]
Specificity: 98.90% [95% CI: 96.84%, 99.77%]

CI, confidence interval.

## Data Availability

The data presented in this study are available on request from the corresponding author. The data are not publicly available due to the protection of personal information of participants in this study.

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
