# Peer review of "Real-Time PCR-Based Screening for Homozygous SMN2 Deletion Using Residual Dried Blood Spots"

_genes, 2023, doi:10.3390/genes14122159_

Round 1
Reviewer 1 Report
Comments and Suggestions for Authors
The SMN2 gene is a recognized modifier of spinal muscular atrophy (SMA). Here, the authors estimated that the frequency of homozygous SMN2 deletion was ~1 in 20 in Japan. They established a real-time polymerase chain reaction (PCR)-based screening method using residual dried blood spots to identify infants with homozygous SMN2 deletion.
Overall, the study is still interesting. However, it needs to be revised.
Major Comments:
1. Please provide a diagram illustrating the position of the primer on the gene.
2. The authors need to discuss more about the advantages and disadvantages of the method in relation to existing methods.
Minor comments:
In Method 2.1, the first and second paragraphs are partially duplicated.
Author Response
(General comments)
Overall, the study is still interesting. However, it needs to be revised.
Answer to (General comments): Thank you very much for your words of encouragement. We revised and improved this manuscript according to your suggestions.
(Major Comments)
- Please provide a diagram illustrating the position of the primer on the gene.
Answer to (Major Comment 1): Thank you very much for your suggestion. We added a diagram illustrating the positions of the primers on the gene in the Methods section (Figures 1 and 2 in the revised version).
- The authors need to discuss more about the advantages and disadvantages of the method in relation to existing methods.
Answer to (Major Comment 2): Thank you very much for your suggestion. Following your suggestion, we added some description about comparison between real time PCR methods, intercalating dye method and fluorescent probe method in the discussion section. We added the following sentences in the Discussion section.
“There are two main methodologies for analysis using real-time PCR: intercalating dye and fluorescent probe methods. In the intercalating dye method, it is comparatively easy to find proper PCR conditions because the only factor to consider is the sequences of primers that anneal to the gene of interest. In addition, the intercalating dye is not expensive. The weakness of the intercalating dye method is that it cannot test multiple genes at the same time.
To the contrary, in the fluorescent probe method, multiple genes are tested at the same time. However, it is difficult to find proper conditions under which many primers and fluorescent probes work well in one tube. In addition, the fluorescent probes are expensive. The more probes there are, the more it costs.
In this study, we adopted the intercalating dye method, because the target gene was only SMN2. However, determining the proper annealing temperature required trial and error.”
Minor comments:
In Method 2.1, the first and second paragraphs are partially duplicated.
Answer to (Minor Comment): Thank you very much for your pointing out our careless mistakes. We omitted the duplicated parts in the revised version.

Reviewer 2 Report
Comments and Suggestions for Authors
In the current manuscript, the authors have detected the SMN2 deletion by PCR RFLP and RTqPCR from the dried blood spot. The authors have performed the study on 300 samples. The authors concludes that by RFLP and RTqPCR methods SMN2 homozygous deletion can be detected. The authors have also discuss the demerits of the study. However, there are a lot of caveats in the study which needs to be addressed. My comments are provided below
1. The introduction is very lengthy and didnot provide any information about the sequences of restriction enzyme sites of the locus of SMN2.
2. The authors should provide a schematic diagram of the SMN2 sequence and enzyme sites with expected product size.
3. After the digestion with restriction enzyme, two bands are expected. If the PCR product amplifies SMN1 and SMN2, then 3 bands are expected after digestion. But the result showed only two bands. The authors defined the lower band as SMN2 after digestion. The product size after digestion should be presented in the text.
4. The RTqPCR result is still not convincing. the authors should present a schematic diagram of binding of primers to the target with sequence. There is still a big difference in Ct value between the two samples red and orange which express the SMN2. With such a variations in Ct value, the RTqPCR methods will make the differentiation inconclusive in unknown samples. Again the RTqPCR was performed with intercalating dye which is a demerits in itself. What is the base line for detecting the amplicon in such a RTqPCR studies? Is it specific for SMN1 or SMN2?
Comments on the Quality of English LanguageThere are a lot of mistakes in the text. Lines 135- 145 are the duplication of previous paragraph.
Author Response
(General comments)
There are a lot of caveats in the study which needs to be addressed. My comments are provided below.
Answer to (General comments): Thank you very much for your kind comments to improve our manuscript. We could revise and improve this manuscript based on your comments.
(Major Comments)
- The introduction is very lengthy and did not provide any information about the sequences of restriction enzyme sites of the locus of SMN2.
Answer to (Major comment 1): Thank you very much for your suggestions. We omitted several sentences from the Introduction section of the revised version, and we added a diagram illustrating the positions of the restriction enzyme sites in the Methods section (Figure 1).
- The authors should provide a schematic diagram of the SMN2 sequence and enzyme sites with expected product size.
Answer to (Major comment 2): Thank you very much for your suggestions. We added a diagram illustrating the position of the enzyme site with expected product size (Figure 1 in the revised version).
- After the digestion with restriction enzyme, two bands are expected. If the PCR product amplifies SMN1 and SMN2, then 3 bands are expected after digestion. But the result showed only two bands. The authors defined the lower band as SMN2 after digestion. The product size after digestion should be presented in the text.
Answer to (Major comment 3): Thank you very much for your insightful and warm comments and suggestions that helped us improve the manuscript. Following your comments and suggestions, we added the following sentences in the Results section.
“The PCR fragment size of SMN2 before Dra I digestion is 187 bp. Digestion with Dra I produced two fragments, 163 bp and 24 bp in size. Because the small 24 bp fragment is difficult to see clearly on agarose gels, previous studies, including van der Steege et al. (Reference number 20) and Srivastava et al (Reference number 8), have used only the large 163 bp fragment as clear evidence for the presence of SMN2. Thus, we show only the large 163bp fragment in the gel electrophoresis picture.”
- The RTqPCR result is still not convincing. the authors should present a schematic diagram of binding of primers to the target with sequence. There is still a big difference in Ct value between the two samples red and orange which express the SMN2. With such a variation in Ct value, the RTqPCR methods will make the differentiation inconclusive in unknown samples. Again the RTqPCR was performed with intercalating dye which is a demerits in itself. What is the base line for detecting the amplicon in such a RTqPCR studies? Is it specific for SMN1 or SMN2?
Answer to (Major comment 3): Thank you very much again for your insightful and warm comments. Your comments here consist of three parts; we answer to each of them below.
(1) Following your comments and suggestions, we added the schematic diagram of binding of primers to the target gene (Figures 2, S1 and S2).
In the revised version, we also added the sequencing results of a false negative sample in the supplementary materials (Figures S1, S2). This sample suggested SMN2-specific primers can amplify SMN1 sequence by mis-annealing to SMN1 sequence under some conditions including low annealing temperature.
(2) Regarding the Ct values, its variation may be inevitable because Ct values are dependent on the DNA amounts in the circles punched from DBS samples. The circles punched from DBS may have varying amounts of DNA. Therefore, we completely agree with your comment, “with such a variation in Ct value, the RTqPCR methods will make the differentiation inconclusive in unknown samples.” Thus, in the real clinical spots, the individuals who had inconclusive DBS samples should be retested in the hospital, collecting fresh blood. Following your comments and suggestions, we added the following sentences in the “Limitations” of the Discussion section.
“Our genetic testing with a real-time PCR method using DBS is just a screening of the disease. Genetic testing using DBS sometimes may produce inconclusive results. Therefore, in the real clinical spots, it is necessary for us to refer the individuals who had inconclusive DBS samples to the expert doctors for the exact diagnosis.”
(3) The base line for detecting the amplicon from the SMN2 gene was obtained after ROX correction of well-to-well variation caused by pipetting inaccuracies and fluorescence fluctuations (Please see the Methods section).
- Comments on the Quality of English Language: There are a lot of mistakes in the text. Lines 135- 145 are the duplication of previous paragraph.
Answer to (Major comment 4): Thank you very much again for pointing out our mistakes. We omitted the duplicated parts in the revised version, and checked and corrected errors in the English language.
Round 2
Reviewer 1 Report
Comments and Suggestions for Authors
Authors have addressed all my comments.
Reviewer 2 Report
Comments and Suggestions for Authors
The authors have addressed all my concerns in the revised version of the manuscript. I support the publication of the manuscript.